# Digital inclusive finance and asset allocation of Chinese residents: Evidence from the China Household Finance Survey

**Kun Li**[1]*, **He Mengmeng**[2], **Junjun Huo**[2]

1 School of International Trade and Economics, Shandong University of Finance and Economics, Jinan, China, 2 School of Economics, Zhejiang University, Hangzhou, China

* Jerry18963262025@163.com

**Data Availability Statement:** All relevant data are within the paper and its Supporting Information files.

**Funding:** The author(s) received no specific funding for this work.

## Abstract

Combined with the expected utility theory, this paper constructs a theoretical analysis framework including the development level, financial literacy, and intelligence level of Inclusive Finance, puts forward the hypothesis of the development of digital Inclusive Finance on household asset allocation, and uses the data of China's household finance survey to verify the theory proposed in this paper. The empirical results show that: (1) digital inclusive Finance can significantly improve the allocation proportion of household risk assets, promote the rational participation of households in the risk financial market, and improve the allocation efficiency of household resources. (2) Digital inclusive finance can significantly improve the income level of family financial investment and optimize family investment decision-making.

## I. Introduction

At the level of financial supply, inclusive Finance can alleviate financial exclusion, reduce the cost of financial services and broaden the scope of financial services using technologies such as big data and cloud computing [1–3]. Many inclusive financial products have emerged that have changed how financial services are delivered and altered people's financial service demands [4, 5]. For example, Alipay provides an integrated financial service platform that combines mobile payment, financial services, credit services, and insurance, reducing people's dependence on traditional financial institutions and enabling more and more households to enjoy financial services at a meager cost. In addition, the development of technology has created a new demand for financial services and lowered the threshold of access to financial services, which makes asset management no longer an exclusive service for high-quality customers, but also allows ordinary people to enjoy professional financial services, fully reflecting the meaning of universal access [6, 7].

At the level of financial needs, the current asset allocation of Chinese households is not optimistic [8–10]. 2019 Annual China Household Wealth Survey Report shows that Chinese households' financial assets are mainly invested in products with high liquidity, security, and low returns. Household cash and bank deposits account for more than 90%, with fewer asset

**Competing interests:** The authors have declared that no competing interests exist.

types and a homogeneous structure that prevents wealth preservation through diversified investments [11]. Due to Chinese households' high-risk aversion and low financial literacy, Chinese families have a low participation rate in the stock, fund, and bond markets. The problem of "limited participation" is severe, and households cannot reasonably use financial markets to achieve wealth preservation and appreciation [12, 13]. At the same time, data from the China Household Finance Survey 2017 show that more than 30% of households are involved in private lending practices. This proportion exceeds 40% in rural areas, suggesting that "over-participation" in informal financial markets is widespread among Chinese households [14, 15].

Against the above background, questions such as how to guide Chinese households to make reasonable use of risky financial markets for rational allocation of household financial assets and how to regulate the economic behavior of Chinese families need to be addressed. Drawing on previous research results, this paper investigates the impact of digital inclusive finance development on the effectiveness of household investment portfolios and the mechanisms, using CHFS data and Peking University Digital Inclusive Finance Index. Compared with existing studies, this paper considers the impact of digital inclusive finance development on investment returns so that this study has both breadth and depth of investment; it analyzes the mechanism of the effects mentioned above and fully considers population heterogeneity; it uses municipal-level data, which improves data accuracy and makes the conclusions more robust compared with previous studies using provincial-level data. In addition, the development of inclusive digital Finance in China has strong national characteristics. In terms of financial services, there is a large gap between China's financial market and that of developed countries, with insufficient coverage of traditional financial services, a large geographical gap between rich and poor, and uneven financial literacy among residents. At the technological development level, digital technology has enabled financial inclusion to penetrate all aspects of people's daily lives and closely integrate with real-life needs. China has adopted a structural relaxation policy and strict regulation, which has put inclusive digital Finance into the fast lane of development. The phenomena mentioned above determine that foreign research results have limited significance for China, and therefore require in-depth research on the impact of the development of inclusive Finance in China's national conditions.

The possible innovations of this paper are: (1) the impact of the level of development of inclusive digital Finance on the effectiveness of household asset allocation is investigated in two dimensions: the risky asset ratio and investment returns. At present, scholars generally focus on the macro impact brought about by the development of inclusive Finance; However, there is literature on the micro impact brought about by inclusive Finance; most of the studies focus on the types of household assets and the proportion of risky assets, and less on the impact of the development of inclusive digital Finance on the level of household investment returns. This paper argues that access to rescues reflects investment efficiency, and it is necessary to continue research on the level of household investment returns. (2) based on considering population heterogeneity, the mechanism of the impact of inclusive digital Finance on household behavior is proposed and verified. Most of the existing empirical studies have been conducted from an econometric perspective and have failed to clarify the transmission mechanism of financial inclusion development on household financial investment behavior. This paper argues that explaining the impact mechanism of financial inclusion and the differences in the impact on different groups of people can help formulate targeted economic policies, help China improve its financial service system, and enhance the degree of financial inclusion in China.

The contribution of this paper and its significance to the real world are shown in the following aspects. In terms of academic importance, the research results of this paper can supplement the existing micro-level impact research and mechanism research of digital Inclusive Finance.

Based on the previous research results and chefs' data, this paper studies the impact and mechanism of the development of digital Inclusive Finance on the effectiveness of household portfolios. Compared with existing studies, this paper considers the result of the development of digital Inclusive Finance on investment returns so that this study has both investment breadth and investment depth. This paper analyzes the mechanism of the above impact and fully considers the heterogeneity of people. This paper uses municipal data, which improves the accuracy compared with previous studies using local data, making the conclusion more robust. In addition, the development of China's digital Inclusive Finance has solid national characteristics. In terms of financial services, there is a large gap between China's financial market and that of developed countries, such as insufficient coverage of traditional financial services, a large regional gap between the rich and the poor, and uneven financial literacy of residents. At the level of technological development, the progress of digital technology has made Inclusive Finance go deep into all aspects of people's daily life and closely integrate with the needs of real life. In terms of policy, China has adopted a policy of structural relaxation while strict supervision, which has brought digital inclusive Finance into the fast lane of development. The above phenomenon determines that the reference significance of foreign research results to China is limited. Therefore, it is necessary to conduct in-depth research on the impact of China's national conditions on the development of Inclusive Finance

In terms of practical significance, the research results of this paper help optimize household investment decision-making. In 2020, China's task of comprehensively eradicating poverty will be achieved, the amount of family investment will increase, and there is a massive demand for wealth management. However, the development of China's household financial market is not perfect. The "limited participation" in the traditional financial market and "excessive participation" in everyday financial need and the household portfolio lacks rationality and effectiveness. How to guide Chinese families to make rational use of the risk financial market, optimize investment decisions, increase family economic well-being, and give full play to the advantages of Inclusive Finance to improve the financial market has become a problem that can not be ignored.

In terms of policy significance, the research results of this paper will help the government formulate accurate and effective inclusive financial development policies and make practical contributions to China to seize the opportunity of targeted poverty alleviation and solve the problem of farmers' poverty. Inclusive Finance helps expand financial coverage, break through the limitations of traditional economic space, and has an essential impact on alleviating Financial Exclusion, reducing the poverty rate, and narrowing the income gap between urban and rural areas. Furthermore, through heterogeneity analysis, this paper explores the differences of different groups affected by the development of Inclusive Finance so that the government can more effectively identify financial assistance groups and improve the accuracy of China's inclusive financial services and the effectiveness of Inclusive Finance-related policies.

## II. Literature review

### 1. Macro impact study on financial inclusion

As early as the last century, [16] suggested that the Internet could alleviate the information asymmetry problem in traditional financial markets, reduce transaction costs and thus facilitate the rapid development of financial markets. [17] conducted an empirical study using data provided by banking regulators and found that an inclusive financial system is essential for stable social development. Its absence will lead to problems such as unequal income distribution and slowing economic growth. Based on this, [18] conducted a study for the financial situation in the Bangladesh region, which confirmed the above findings and quantitatively measured

the relationship between financial inclusion and GDP. [19] found that financial inclusion can effectively address poverty and reduce income disparity by analyzing data from developed and developing countries. [20] also reached similar conclusions. [21] pointed out that inclusive Finance can help low-income people to access funds, thus promoting personal asset accumulation and improving household status, and should be placed at the core of financial development. [22] compared the availability, utilization rate, and depth of use of financial services in different countries and concluded that inclusive Finance has a role in poverty reduction, but there are currently significant regional differences in the development of inclusive Finance, for which efforts should be made to eliminate regional imbalances and develop inclusive Finance globally to achieve sustainable financial development globally. [23] conducted a study on financial inclusion in India and concluded that increased financial inclusion could contribute to national economic growth. [24] surveyed farmers in Kenya and found that inclusive digital Finance can help farmers access financial services by improving payment convenience and providing access to subsidies, providing new channels and methods to address national poverty.

## 2. Micro impact study on financial inclusion

At the micro-level, the development of inclusive Finance can also rapidly reduce the cost of financial services. Currently, inclusive Finance is innovative in payment and settlement, investment and Finance, credit business processing, and supplying financial services, which can significantly improve traditional financial institutions' service efficiency and enable new financial services to benefit the public.

In terms of asset allocation, [25] studied the asset allocation of farm households using CHFS data and showed that inclusive financial development could promote financial market participation and increase the share of risky asset allocation among farm households. [26], from the perspective of household financial literacy, argues that inclusive financial development can broaden people's financial literacy and lead to a more rational allocation of household financial assets. This phenomenon is more pronounced among low-income households and rural households. [27] also concluded that digital financial inclusion has a positive impact on household asset allocation using CLDS data.

In terms of residential consumption, there have been numerous studies showing that alleviating financial constraints can impact the consumption behavior of residents [28]. A study by [29] using CFPS and the North University Inclusive Finance Development Index concluded that inclusive digital Finance could effectively alleviate household liquidity constraints, improve payment convenience, and release consumption demand. [30] found that the development of inclusive digital Finance can significantly promote the upgrading of the consumption structure of rural residents. [31]conducted an empirical analysis for provincial panel data in China. She found that the development of inclusive digital Finance can significantly narrow the consumption gap between urban and rural areas and help to further promote China's rural revitalization strategy.

The impact of digital financial inclusion development on enterprises is more about improving the efficiency of financing and improving the level of financial stability. [32] Internet finance can help enterprises obtain a more comprehensive range of financial support, improve their financing efficiency and reduce their financing costs, and [33] argue that new technologies can help match the supply and demand of funds over long distances and improve the efficiency of capital use. The development of information technology also helps overcome the information asymmetry problem in traditional financial markets. It enables financial institutions to improve the accuracy of enterprise risk assessment while reducing assessment costs

and helping small businesses obtain financing with greater efficiency [34]. [35] found that the development of financial inclusion can improve the financial stability of online merchants and enhance their resilience to business shocks by analyzing the data of "Ant Financial Services."

## 3. Research on factors influencing household asset allocation

Life cycle theory suggests that the allocation of household financial assets is often related to the life cycle of family members and that the lifetime utility of family members can be maximized through the proper distribution of financial support. Therefore, household asset allocation is often influenced by household expectations of future income and expenditure. Many studies have focused on the impact of background risk on household financial risk asset allocation. Foreign scholars have found that households with uncertain future income increase the share of illiquid household assets and reduce the risky investments invested [36, 37]. [38] found that income risk, health risk, and liquidity risk hurt risky asset investments when studying U.S. household holdings, and this finding has been confirmed by other scholars [39]. However, [40], analyzing Dutch savings data from 1993 to 1998, find that income uncertainty does not affect household financial asset investment behavior. [41] by analyzing data on French workers, households with high-income luck instead increase their share of risky asset investments. [42], analyzing US SCF data finds that credit constraints reduce the percentage of households' risky asset holdings.

In terms of household characteristics, many studies have focused on the impact of household property holdings on household financial behavior. [43] argues that household property status significantly affects household stock market participation, with the property having high switching costs and being less liquid compared to other assets. The higher the proportion of property in total household assets, the lower the proportion of household stock assets. [44] also concludes a significant crowding-out effect of property on investment in risky financial help based on Chinese investor data.

Contrary to these findings, [45], studying the financial behavior of Italian households, finds that owning property instead helps residents obtain mortgage credit and thus promotes risky investments. [46] found significant heterogeneity in asset allocation across income levels, with lower-income households in the U.S. prefer to hold cars and more liquid assets, the middle class choosing to own property, and the rich picking to make investments. [47] argue that household wealth levels can increase participation in formal financial services and that this effect is more pronounced in rural areas.

## 4. Review of the literature

Foreign financial markets developed earlier, and the system is relatively perfect. Foreign scholars have more comprehensive and rich research on family financial asset allocation. There are still some problems in China's financial market compared with foreign countries, such as unreasonable investment structure and unreasonable financial product structure. In addition, there are still some problems in China, such as farmers' poverty complex and expensive financing for small enterprises. These difficulties faced by traditional Finance provide space for developing Inclusive Finance in China. Compared with family finance, inclusive Finance has been created for a short time,

At present, the empirical research on Inclusive Finance in China mainly focuses on the macro impact of Inclusive Finance, such as promoting economic growth and narrowing the income gap between urban and rural areas. The main research body is mainly traditional financial institutions (banks, insurance companies, etc.). On the other hand, foreign research covers a wide range, primarily focusing on the fact that inclusive Finance can reduce financial

exclusion and eliminate economic discrimination, which impacts economic development and social stability.

Although scholars have conducted various studies on the impact of Inclusive Finance, there are still the following deficiencies:

1. Existing studies generally pay attention to the macro impact of the development of Inclusive Finance, less research on the micro implications of Inclusive Finance, and less empirical literature on the relationship between digital finance development and residents' financial management and investment. According to the statistics of Bain consulting in 2018, China has more than 190 trillion yuan of personal investable assets and has a massive demand for wealth management. However, China's household asset allocation is mainly bank deposits and real estate, which lacks rationality and effectiveness. Therefore, this paper believes that the research on the impact of the development of digital Inclusive Finance on household financial asset allocation is fundamental.

2. Most of the existing empirical studies are from measurement but fail to clarify the transmission mechanism of inclusive financial development to family economic behavior. This paper believes that defining the impact mechanism of Inclusive Finance and its impact on different groups will help to formulate targeted monetary policies, help China improve the financial service system, and enhance the degree of financial inclusion in China.

3. Foreign research covers a wide range. However, due to the rapid development of the Internet in recent years, there is a large gap between China's financial market and developed countries. Furthermore, due to the insufficient coverage of traditional financial services, the large gap between the rich and the poor in China, the uneven financial literacy of residents, and other factors, the reference significance of foreign research results to China is limited; it is necessary to conduct in-depth research on the impact of China's national conditions on the development of Inclusive Finance.

## III. Theoretical analysis and hypothesis

Based on the expected utility theory and asset allocation framework, this paper analyzes the effectiveness of digital inclusive financial development on household asset allocation at the theoretical level. Assuming that households make financial investments in period t and receive investment returns in period $t + 1$, the corresponding household utility functions and the household constraints considering the development of inclusive digital Finance can be expressed as follows.

$$\max \frac{E_t(c_{t+1}^{1-\gamma})}{1-\gamma} \tag{1}$$

$$s.t \ \ C_{t+1} = (1 + R_{p,t+1})W_t + L_{t+1} - (T_t - w \cdot \delta \cdot kn_t) \tag{2}$$

where $C_{t+1}$ denotes the consumption expenditure of the household in period $t + 1$, $\gamma$ denotes the relative household risk aversion factor, $R_{p,t+1}$ denotes the investment portfolio obtained by the household in period $t + 1$, $W_t$ denotes the household's wealth level in period $t$, $L_{t+1}$ denotes the household's labor income in period t + 1, $T_t$ Denotes the financial transaction cost without considering the development of inclusive digital Finance and $w·\delta·kn_t$ denotes the financial transaction cost reduced by the development of inclusive digital Finance. Digital financial inclusion can have an impact on household financial literacy and intelligence levels, which is reflected in the formula $w$ measuring the level of household financial literacy, $\delta$ measuring the

level of household intelligence, $n_t$ represents the level of development of digital inclusion, and $k$ represents the conversion factor between the development of digital inclusion and financial transaction costs.

In this paper, we assume that the household will choose to invest in risky financial assets and risk-free financial assets in period $t$. Therefore, their shares in the household's investable amount $(W_t + L_t - T)$ are $\hat{a}_t$, and $1 - \hat{a}_t$ The intertemporal return on risky investments $R_{t+1}$ is assumed to follow a positive-terms distribution with mean $E_t R_{t+1}$ and variance $\sigma_t^2$. The level of return on risk-free financial assets is assumed to be the corresponding return level is $R_{f,t+1}$, then the household portfolio return can be expressed as follows.

$$R_{p,t+1} = \hat{a}_t R_{t+1} + (1 - \hat{a}_t) R_{f,t+1} \tag{3}$$

Solving the above equation yields: $\hat{a}_t = \frac{E_t R_{t+1} - R_{f,t+1} + \sigma_t^2/2}{\gamma \sigma_t^2}$;

Then the household risk asset investment ratio can be expressed as:

$a = \frac{\hat{a}_t}{W_{t+1}} [W_t + L_t - (T_t - w \cdot \delta \cdot k n_t)]$.

Derivation of the above equation for the level of development of digital financial inclusion reveals that: $\frac{da}{dn_t} \geq 0, \frac{dR_{p,t+1}}{dn_t} \geq 0$. This suggests that the development of digital financial inclusion can positively impact household financial asset allocation, and based on the above calculation results, this paper proposes the following two hypotheses.

**Hypothesis 1: Digital financial inclusion development can increase the proportion of household risk assets allocated.**

**Hypothesis 2: The development of inclusive digital Finance can improve return on household asset allocation.**

The traditional financial theory assumes investors to be well-informed and rational in their decisions, but these two assumptions are often difficult to hold in real life. Some scholars have incorporated financial literacy into the analytical framework of asset allocation to make the study more relevant to the actual financial decision-making situation [48, 49]. [50] studied the relationship between household financial asset allocation and financial literacy regarding investment diversification, risk-taking level, and equity investment. They concluded that increased financial literacy effectively improves household investment efficiency. [51], using US SCF data, concluded that financial literacy has a human capital effect and can increase households' net wealth. In addition to this, a large body of research demonstrates that financial literacy increases household investment [52]. The development of technology has broadened people's access to information and lowered the threshold of access to information. It has been demonstrated positively that Finance combined with new technology is consistent with the classical technology diffusion theory [53], allowing people to quickly access new products and attract investors to invest in them. Therefore, this paper proposes hypothesis 3.

**Hypothesis 3: Digital financial inclusion development can influence household economic behavior by affecting financial literacy, and this effect varies across income groups and age groups.**

The development of Internet technology has made financial services no longer confined to time and place, which can effectively reduce market friction and transaction costs. The emergence of Internet finance has also changed the service model of traditional Finance, showing the qualities of "decentralization" and "financial disintermediation" [54]. The emergence of Internet finance has also changed the service model of traditional Finance, showing the qualities of "decentralization" and "financial disintermediation" [54]. Thus, on the one hand, the

development of inclusive Finance relies on promoting Internet technology. On the other hand, the development of inclusive Finance also promotes people's access to new financial service models, increases the frequency of people's use of Internet finance, increases people's willingness to use new financial products, especially Internet financial products, and enables people to realize financial product transactions more conveniently and quickly and enjoy the convenience brought by the development of Internet technology. Accordingly, this paper puts forward hypothesis 4.

**Hypothesis 4: The development of digital inclusion can affect household financial behavior by influencing the willingness of households to use the Internet.**

## IV. Study design

### 1. Data source

China Household Finance Survey (CHFS) is a nationwide sampling project on micro information of household finance. The database adopts a stratified, three-stage, and PPS sampling method, with a low rejection rate and data close to the census results, which is representative and can facilitate research on household financial behavior and macro policy formulation in China. The CHFS data covers information on household housing assets, financial finances, liabilities and credit constraints, income consumption, social security, payment habits, and demographic characteristics, providing a comprehensive and detailed picture of household economic behavior.

The CHFS database is open for applications for 2011, 2013, 2015, 2017, and 2019 survey results data. This paper uses data from five rounds of survey results from 2011–2019, whose sample covers 29 provinces (including autonomous regions and municipalities directly under the central government), 355 counties (including districts and county-level cities), and 1428 village (residence) committees, with a total sample of 140011 households. Since the original CHFS data contained a large number of missing values and outliers (e.g., household income, household assets, personal risk attitudes, etc.), to ensure the reliability of the results, this paper eliminated samples containing missing values and outliers for critical variables (e.g., household income, household assets, etc.), and obtained a total of 118,784 valid samples.

The Digital Inclusive Finance Index of Peking University is compiled by the Digital Inclusive Finance Research Center of Peking University and Ant Financial Services based on Alipay transaction account data. Including municipalities directly under the Central Government and autonomous regions), 337 cities above prefectural level (including autonomous regions, leagues, etc.), and about 2,800 counties (including county-level towns, municipal districts, etc.), which can reflect the dynamic process of digital inclusive finance development in each province, city, and county. This paper uses the data of the Peking University Digital Inclusive Finance Index at the municipal level.

### 2. Variable selection

This paper defines the effectiveness of household asset allocation in two dimensions: risky financial market participation and investment returns. It uses the proportion of risky household assets to measure difficult financial market participation. There is a distinction between the broad and narrow sense of household risky financial assets. The general sense of risky assets includes foreign exchange investment, real estate investment, gold investment, futures investment, borrowing situation, etc. The narrow sense of risky household assets refers only to the holdings of household funds, bonds, and stocks. Given data availability, this paper selects the participation of sample households in the fund, cement, and stock markets as the participation in household risky financial markets.

The data relating to financial products were obtained from the relevant responses in the CHFS2017 financial assets section questionnaire, and some samples did not answer the specific amount of investment funds, bonds, and stocks, and only a range of investment amounts was provided in the questionnaire; to reduce the sample loss, the median was taken as a substitute for the specific investment amount for this part of the sample in this paper. Since the purpose of this paper is to examine whether households can diversify their investments and hedge their risks through financial markets, the ratio of risky financial assets to total household assets is used to measure participation in complex financial needs.

The explanatory variable of this paper is the Digital Inclusive Finance Index, which comes from the city-level data of the BYU Digital Inclusive Finance Index, with 337 city-level data covering the overall development level, coverage, and depth of use of inclusive digital Finance, with secondary indicators covering the application of inclusive digital Finance in cities in terms of investment, payment, credit, credit collection, insurance, money fund use, and digital development. Thus, the index can provide an all-around portrayal of the level of development of inclusive digital Finance in cities.

Referring to previous studies, at the individual level, this paper selects household head education, marriage, gender, age, personal risk attitude, and attention to financial information as control variables. Among them, the risk attitude data were obtained from the responses to the investment choice questions in the financial literacy section of the CHFS 2017 questionnaire, defining users who are unwilling to take risks and willing to accept slightly lower risks as risk-averse, those who prefer high-risk and high-return projects as risk-averse, and those in-between as risk-neutral. The financial information concern data were obtained from respondents' responses to the CHFS questionnaire on the degree of concern for financial information. In this paper, 1 is defined as never concerned, and 5 is defined as very concerned. At the household level, this paper selects the log of household assets, the log of household income level, the proportion of children, the proportion of older adults, the health level of household members, the presence of a financial-related practitioner, and the household commercial insurance status as control variables, where the household financial practitioner variable is a binary variable of 0 and 1. If someone in the household is engaged in the financial industry or was engaged in the financial industry before retirement, it is defined as 1. Otherwise, it is taken as 0. To control the influence of regional macro factors on household economic behavior, the level of GDP per capita, GDP growth rate, urbanization rate, and urban economic development are selected as city-level control variables in this paper. Financial literacy is a combination of financial knowledge, behaviours and skills possessed by an individual, and measures the financial knowledge reserve possessed by an individual and the ability to apply financial knowledge to solve practical financial problems. This paper uses respondents' responses to the financial literacy questions in the questionnaire as the basis for determining the level of financial literacy.

## 3. Model construction

In studying the impact of the development of inclusive digital Finance on the proportion of household financial risk holdings, this paper selects the proportion of household risky financial assets holdings finance to total assets $riskasset_i$ as the explanatory variable. Since the ratio of risky investments $riskasset_i$ is mainly a 0–1 variable, this paper uses the Tobit model for estimation, and the model is set as follows.

$$riskasset_i = \alpha_1 fin_i + \beta_1 X_i + \mu_i \tag{4}$$

where $fin_i$ represents the local financial inclusion index and $X_i$ defines the control variables, including individual, household, and city-level characteristics.

In studying the impact of digital financial inclusion development on household investment returns, the Sharpe ratio *sharp ratio*$_i$ is chosen as the explanatory variable. However, for households that do not participate in investment in risky financial assets, their corresponding Sharpe ratios are 0. Therefore, Sharpe ratios are mostly 0–1 variables, which are also estimated using Tobit models, and the related regression models are as follows.

$$sharpratio_i = \alpha_2 fin_i + \beta_2 X_i + \mu_i \tag{5}$$

This paper argues that digital financial inclusion can affect household economic behavior by influencing the level of financial literacy and the level of Internet usage. The analysis for Mechanism 1 can be divided into two steps: the first step tests the impact of financial inclusion development on financial literacy, and the corresponding regression model follows.

$$knowledge_i = \alpha_1 fin_i + \beta_1 X_i + \mu_i \tag{6}$$

To fully consider the population heterogeneity, this paper adds the asset interaction term and the age group interaction term to the above model to explore the differences in the impact of digital financial inclusion development on the financial literacy of different population groups. The corresponding regression models are as follows.

$$knowledge_i = \alpha_1 fin_i + \beta_1 asset_i + \beta_2 asset_i \times fin_i) + \beta_3 X_i + \mu_i \tag{7}$$

$$knowledge_i = \alpha_1 fin_i + \beta_1\ agec_i + \beta_2(agec_i \times fin_i) + \beta_3 X_i + \mu_i \tag{8}$$

The analysis idea for mechanism two also adopts a similar concept to the analysis of mechanism one. This paper chooses two variables, whether the household uses Internet financial products (using Internet financial products takes the value of 1. Otherwise, it takes the value of 0) and the household's use of Internet (used takes the value of 1. Otherwise, it takes the value of 0), as the willingness to use Internet variables for regression analysis. The corresponding regression models are as follows.

$$internet_i = \alpha_1 fin_i + \beta_1 X_i + \mu_i \tag{9}$$

$$riskasset_i = \alpha_3\ internet_i + \beta_1 X_i + \mu_i \tag{10}$$

$$sharp\ ratio_i = \alpha_4\ internet_i + \beta_2 X_i + \mu_i \tag{11}$$

## V. Results and discussion

### 1. Impact of financial inclusion development on household financial market participation

This paper first estimates the impact of digital inclusive financial development on the proportion of risky assets of households and adds individual characteristic variables, household characteristic variables, and city characteristic variables in turn for regression; Table 1 reports the regression results, which show that the coefficient corresponding to the digital inclusive financial index is significantly positive at the 1% level, indicating that digital inclusive economic development can effectively promote household participation in risky financial markets and enhance household Hypothesis 1 holds. Finally, this paper calculates that the marginal effect corresponding to the digital inclusive finance index in (3) is 0.2349, indicating that for each unit increase in the digital inclusive finance index 0.2349%, with other variables held constant.

**Table 1. Impact of financial inclusion development on the share of household risk assets.**

| Variables | Percentage of financial assets | | |
|---|---|---|---|
| | **(1)** | **(2)** | **(3)** |
| Financial Inclusion Index | 11.265*** | 3.108*** | 2.755** |
| | (9.26) | -3.919 | -2.315 |
| Education | | 0.878*** | 0.880*** |
| | | -7.949 | -7.935 |
| Marriage | | -1.08 | -1.062 |
| | | (-1.46) | (-1.43) |
| Gender | | -1.273*** | -1.259*** |
| | | (-3.88) | (-3.84) |
| Age | | 0.442*** | 0.442*** |
| | | -5.519 | -5.525 |
| Age squared | | -0395** | -0.393** |
| | | (-5.25) | (-5.24) |
| Risk attitude | | 1.755** | 1.753** |
| | | -6.669 | -6.665 |
| Log of Assets | | 0.992** | 0.993** |
| | | -6.529 | -6.545 |
| Income | | 0.784*** | 0.782*** |
| | | -4.539 | -4.545 |
| Proportion of children | | 0.909 | 0.939 |
| | | -0.919 | -0.965 |
| Proportion of elderly | | 1.924*** | 1.916*** |
| | | -3.229 | -3.235 |
| Family Health | | -0.071 | -0.074 |
| | | (-0.12) | (-0.12) |
| Household financial status | | 1.064** | 1.051** |
| | | -2.089 | -2.065 |
| Commercial Health Insurance | | 1.145** | 1.121** |
| | | -2.019 | -1.985 |
| GDP per capita | | | 0.015 |
| | | | -0.115 |
| GDP growth rate | | | 0.082 |
| | | | -0.845 |
| Urbanization Rate | | | 0.046 |
| | | | -0.025 |
| Urban Financial Development | | | -0.181 |
| | | | (-1.57) |
| Adj.R$^2$ | 0.236 | 0.256 | 0.235 |
| Chi2 statistic | 215.26 | 215.75 | 216.46 |
| P-value | 0.000 | 0.000 | 0.000 |

Note: (1) Parentheses indicate the t-values corresponding to the estimated coefficients; (2) *, **, and *** indicate significance at the 10%, 5%, and 1% statistical levels, respectively, as in the following table.

## 2. The impact of financial inclusion development on household financial investment returns

Table 2 reports the impact of the level of development of the digital inclusion finance index on the Sharpe ratio. The regression results show that the development of inclusive digital Finance

**Table 2. Impact of financial inclusion development on household Sharpe ratio.**

| Variables | Sharpe Ratio | | |
|---|---|---|---|
| | (1) | (2) | (3) |
| Financial Inclusion Index | 0.156* | | |
| | (1.94) | | |
| Payment Index | | 0.056** | |
| | | (2.35) | |
| Money Fund Index | | | 0.158*** |
| | | | (2.61) |
| Adj.R² | 0.268 | 0.246 | 0.285 |
| Chi2 statistic | 1568.26 | 1486.65 | 1258.62 |
| P-value | 0.000 | 0.000 | 0.000 |

helps to improve the level of household investment returns. To explore the impact of different dimensions of the development of inclusive Finance on household investment returns, this paper chooses indicators in the depth of use dimension of the digital inclusive finance index to replace the total index for regression. The empirical results show that the corresponding coefficients of the payment index and the money fund index are significantly positive, indicating that there is a significant positive impact of payment convenience and money fund use Sharpe ratio, suggesting that lowering the threshold of financial service use can significantly increase the level of household investment returns, and hypothesis 2 holds.

The results of the two regressions above show that the gender of the household head significantly affects the financial behavior of the household, with households headed by women holding a higher proportion of risky assets and having a higher level of return on their investment portfolios. Furthermore, the coefficient corresponding to the education level of the household head is significantly positive, which may be because people with higher education levels tend to have better learning ability, better ability to use investment skills, and better timing of investments, resulting in better financial performance. In addition, the age of the household head is significantly positive. In contrast, the squared term of the period is very harmful, indicating that the effect of the age of the household head on the effectiveness of household asset allocation has an inverted U-shape, which may be because before old age, as the age of the household head increases, the experience of the household head increases. As a result, they can make a more reasonable asset. This may be because before old age, as the household head grows older, the household head's experience and investment experience increases, and they can allocate assets more rationally and have higher returns.

## VI. Analysis of impact mechanisms

### 1. Improve family financial literacy

The current widely accepted definition of financial literacy is the combination of knowledge, skills, and behaviors required to make sound financial decisions. However, there is no uniform standard for measuring financial literacy in the process of econometric analysis. Some scholars have studied using education level as a proxy variable for financial literacy; for example, (Starr & Kennickell, 2010) used years of education to measure financial literacy in their study of household deposit insurance in the United States. (Chiarella et al., 2006), on the other hand, argue that there is no simple equivalence between education level and financial literacy. Education focuses more on the knowledge level and financial literacy, emphasizing more on technology application. Since the CHFS questionnaire includes questions on financial literacy, the

**Table 3. Impact of financial inclusion development on financial literacy.**

| Variables | Financial Literacy | | |
|---|---|---|---|
| Financial Inclusion Index | 0.356*** | 1.952*** | 0.305*** |
| | (4.27) | (3.52) | (2.62) |
| Asset interaction term | | -0.156** | |
| | | (-2.63) | |
| Age group interaction | | | 0.268* |
| | | | (1.48) |
| Individual variables | Controlled | Controlled | Controlled |
| Household variable | Controlled | Controlled | Controlled |
| City variable | Controlled | Controlled | Controlled |
| Adj.$R^2$ | 0.162 | 0.156 | 0.178 |
| Chi2 statistic | 2653.54 | 2586.62 | 2468.65 |
| P-value | 0.000 | 0.000 | 0.000 |

correct response rate of the sample to financial questions is chosen as a measure of financial literacy. In this paper, low financial literacy is defined as being unable to answer financial questions or having a correct rate of zero, high financial literacy is defined as answering all questions correctly, and intermediate financial literacy is defined as being in between.

Column (1) of Table 3 reports the impact of the level of digital financial inclusion development on individuals' financial literacy, and the coefficient corresponding to the digital financial inclusion index is significantly positive, indicating that the level of digital financial inclusion development can dramatically improve the level of financial literacy. This paper argues that there are differences in the mechanisms of people's financial behaviors with different characteristics due to the development of inclusive Finance. This paper adds the asset interaction term and the age group interaction term to the above model to fully consider the population heterogeneity. Since the financial market in China differs significantly before and after the reform and opening up, people's cognitive level and investment attitudes also vary, so this paper divides the population into age groups according to whether they were born before the reform and opening up.

The last two columns of Table 3 report the regression results of the interaction terms. The coefficient corresponding to the asset interaction term is significantly negative, which indicates that the development of inclusive digital Finance can increase the financial literacy level of poor people to a greater extent than that of rich people; the coefficient corresponding to the age group interaction term is significantly positive, which indicates that the development level of inclusive digital Finance has a more significant impact on the financial literacy of people born before the reform and opening up.

The above empirical results show that the level of development of inclusive digital Finance can positively and significantly affect the effectiveness of household financial asset allocation by improving financial literacy. Furthermore, after considering population heterogeneity, it is found that inclusive digital Finance can dramatically improve the financial literacy of the group with lower asset levels and born before the reform and opening up and has a more significant impact on the effectiveness of household financial asset allocation of this group, and hypothesis 3 holds.

## 2. Increase willingness to use the Internet at home

(Kelton & Pennington, 2012) found that financial accessibility has a significant impact on household investment behavior. The development of technology has given rise to new service models. Both brick-and-mortar financial institutions and Internet-based financial trading platforms have

**Table 4. Impact of financial inclusion development on willingness to use the Internet.**

| Variables | Internet Wealth Management | | Internet usage | |
|---|---|---|---|---|
| Financial Inclusion Index | 0.456*** | 0.492*** | 0.562*** | 1.325** |
| | (7.56) | (5.35) | (6.22) | (5.29) |
| Individual Variables | Uncontrolled | Controlled | Uncontrolled | Controlled |
| Household variable | Uncontrolled | Controlled | Uncontrolled | Controlled |
| City Variables | Uncontrolled | Controlled | Uncontrolled | Controlled |
| Adj.R$^2$ | 0.256 | 0.458 | 0.354 | 0.368 |
| Chi2 statistic | 56.235 | 52.346 | 56.325 | 52.367 |
| P-value | 0.000 | 0.000 | 0.000 | 0.000 |

become important influences on financial accessibility. In China, the regional distribution of physical, financial institutions varies widely and is an important limiting factor for financial market participation. The development of inclusive Finance relying on the Internet and big data technologies has reduced the dependence of household financial behavior on physical, financial institutions and weakened the avenues for limited participation. This paper argues that the development of inclusive digital Finance can impact household financial behavior by increasing the willingness to use Internet finance to expand the scope of financial services.

Table 4 reports the impact of the level of development of inclusive digital Finance on the willingness to use Internet products. In this paper, regressions are conducted using whether households use Internet financial products and households' use of the Internet and apps, respectively. The regression results show that the development of inclusive digital Finance can enhance households' use of Internet products and improve the availability of Internet finance.

Table 5 reports the effect of willingness to use Internet products on household financial behavior, and the regression results show that the coefficient corresponding to a desire to use the Internet is significantly positive, indicating that the use of the Internet can positively affect the effectiveness of household financial asset allocation, and Hypothesis 4 holds.

## 3. Robustness test

The above findings suggest that financial inclusion significantly contributes to the effectiveness of household financial asset allocation. To ensure the robustness of the results, this paper uses a permutation of indicators and group regressions for robustness testing.

**Table 5. Effect of willingness to use Internet finance on the effectiveness of household asset allocation.**

| Variables | Percentage of risky assets | | Sharpe Ratio | |
|---|---|---|---|---|
| Internet money management | 3.562*** | | 0.264*** | |
| | (7.16) | | (8.16) | |
| Internet use | | 1.165*** | | 0.061*** |
| | | (3.18) | | (3.71) |
| Personal Variables | Controlled | Controlled | Controlled | Controlled |
| Household variable | Controlled | Controlled | Controlled | Controlled |
| City variable | Controlled | Controlled | Controlled | Controlled |
| Adj.R$^2$ | 0.262 | 0.264 | 0.286 | 0.213 |
| Chi2 statistic | 1568.26 | 1456.38 | 1563.28 | 1563.05 |
| P-value | 0.000 | 0.000 | 0.000 | 0.000 |

i. **Switching to different indicators**. This paper argues that the total index of digital inclusive financial development of Peking University has a single dimension of investigation, which cannot reflect the influence of the breadth and depth of digital inclusive economic growth on the effectiveness of household financial asset allocation. Therefore, this paper uses the first-level digital inclusive financial index instead of the total index for regression, and the regression results are shown in Table 6. The coefficients corresponding to the breadth of coverage and depth of use are significant king, indicating that both the breadth and depth of digital inclusive finance development can enhance the effectiveness level of household financial asset allocation, which suggests that the conclusions of this paper are more robust.

ii. **Consider the differences between residents of different households**. This paper uses the household head's household registration to distinguish between urban and rural areas and then examines the differences between urban and rural areas. The regression results in Table 7 show that the estimated coefficients of the financial inclusion index are significantly positive after controlling for individual characteristics variables, household characteristics variables, and urban variables, which indicates that financial inclusion has a significant contribution to the effectiveness of asset allocation for households with different household status. The conclusions of this paper hold robustly.

## VII. Conclusions and recommendations

### 1. Conclusion

This paper studies the impact of the development of digital Inclusive Finance on the effectiveness of household financial asset allocation. Firstly, this paper combs the measurement standards of relevant variables, the relevant research results of Inclusive Finance and household asset allocation, and finds that the current research on the household micro impact of digital Inclusive Finance mostly stays at the level of household asset types and the proportion of household risk assets, without considering the return on household investment; Most of the existing empirical studies fail to clarify the transmission mechanism of inclusive financial development to household financial investment behavior from the perspective of measurement. This study aims to improve the above deficiencies, supplement and improve the existing research results.

This paper combines utility theory to construct a theoretical analysis framework that includes the level of financial inclusion development, financial literacy, and intelligence, and then derives and analyzes a hypothesis about the impact and mechanism of action between the

**Table 6. Switching to a depth of coverage and breadth of use indicators.**

| Variables | Breadth of coverage | | Depth of use | |
|---|---|---|---|---|
| | Percentage of risky assets | Sharpe Ratio | Percentage of risky assets | Sharpe Ratio |
| Breadth of coverage | 1.856** | 0.158* | | |
| | (2.64) | (1.68) | | |
| Depth of use | | | 1.95** | 2.67*** |
| | | | (2.65) | (3.48) |
| Individual variables | Controlled | Controlled | Controlled | Controlled |
| Household variable | Controlled | Controlled | Controlled | Controlled |
| City variable | Controlled | Controlled | Controlled | Controlled |
| Adj.R$^2$ | 0.268 | 0.246 | 0.281 | 0.215 |
| Chi2 statistic | 1563.26 | 1586.35 | 1563.56 | 1567.25 |
| P-value | 0.000 | 0.000 | 0.000 | 0.000 |

**Table 7. Robustness tests considering differences between urban and rural residents.**

| Variables | Breadth of coverage | | Depth of use | |
|---|---|---|---|---|
| | Percentage of risky assets | Sharpe Ratio | Percentage of risky assets | Sharpe Ratio |
| Financial Inclusion Index | 6.254** | 0.345** | 3.628** | 0.628* |
| | (2.36) | (2.68) | (1.26) | (1.16) |
| Individual Variables | Controlled | Controlled | Controlled | Controlled |
| Household variable | Controlled | Controlled | Controlled | Controlled |
| City variable | Controlled | Controlled | Controlled | Controlled |
| Adj.$R^2$ | 0.265 | 0.368 | 0.246 | 0.325 |
| Chi2 statistic | 263.25 | 246.35 | 246.85 | 265.89 |
| P-value | 0.000 | 0.000 | 0.000 | 0.000 |

level of digital financial inclusion development and household asset allocation. This paper argues that the development of inclusive digital Finance can enhance the effectiveness of household asset allocation, and this effect is carried out by improving financial literacy and residents' willingness to use the Internet. The mechanism of action differs among different income and age groups, and the effect is more pronounced in older age groups with lower asset levels.

In terms of empirical evidence, this paper defines the effectiveness of household asset allocation in terms of two dimensions: household risk market participation and investment returns, and studies investment breadth and depth. This paper selects CHFS data and BYU Digital Inclusive Finance Index to test the hypotheses proposed in this paper. The empirical results show that (1) the development of inclusive digital Finance can significantly increase the proportion of household risky asset allocation and promote households' reasonable participation in the complex financial market to improve the allocation efficiency of household resources. (2) The development of inclusive digital Finance can significantly improve the return level of household financial investment and optimize household investment decisions. Finally, this paper analyzes the mechanism of the above findings. It concludes that: (1) the development of inclusive digital Finance can improve residents' financial literacy, enhance households' knowledge about financial product selection and financial risk management, improve the efficiency of financial decision-making, and thus influence households' financial behavior. And the mechanism has a more substantial impact on people with lower income and older age. (2) The development of digital inclusive financial development can fully play the advantages of Internet technology, increase residents' willingness to use the Internet, and enable them to enjoy the dividends of technological development, thus optimizing household financial market participation.

## 2. Recommendation

Based on the above research findings, this paper proposes the following policy recommendations.

1. Play a leading role of the government and build a suitable environment for developing inclusive digital Finance. Digital inclusive Finance can break the current financial dilemma, help implement the rural revitalization strategy, and realize the leap from precise poverty alleviation and poverty eradication to precise policy-making. However, the development of inclusive digital Finance needs to rely on a good development environment. Therefore, the government should focus on improving the infrastructure construction and credit system to develop digital inclusive Finance to promote the complementary advantages of digital

inclusive finance-related parties and form an excellent financial, ecological pattern of win-win cooperation.

2. Focusing on financial information regulation and regulating online financial information delivery. This study considers improving financial literacy as one of the mechanisms of digital inclusive Finance's influence on household economic behavior. However, there are still problems in China, such as spreading false financial information, promoting financial products and services banned by relevant departments, and distorting the interpretation of national fiscal and monetary policies and financial management policies, which hinder residents' accurate access to practical information. Therefore, the government should focus on improving the quality of financial information services, punishing illegal and unlawful dissemination of false financial information and wrong financial information, and promoting the orderly development of financial information services.

3. Strengthen financial education and enhance household financial literacy. This paper finds that improving the level of household financial literacy can significantly enhance the effectiveness of household asset allocation. At present, the overall financial literacy of Chinese residents is at a low level. As a result, financial frauds occur occasionally, and most residents are unable to correctly understand financial products and correctly view financial risks. Therefore, the government should focus on increasing financial education, popularizing basic financial knowledge through various channels such as the Internet and television, helping residents establish risk awareness, guiding investors to invest rationally, and promoting the healthy and sustainable development of China's financial market.

4. Give full play to the advantages of advanced technology and standardize and promote financial service innovation under the condition of controllable risk. Financial service innovation can enrich product supply, optimize service quality and improve financial service efficiency. The government should implement the new development concept, comply with the development requirements of the times, and strive to promote the deep integration of Finance and science and technology under the condition of controllable risk, so as to make the achievements of scientific and technological development better benefit the people's livelihood.

## Supporting information

**S1 Dataset.**
(XLSX)

## Author Contributions

**Conceptualization:** Kun Li, He Mengmeng.

**Formal analysis:** Kun Li, He Mengmeng, Junjun Huo.

**Funding acquisition:** Junjun Huo.

**Investigation:** Junjun Huo.

**Methodology:** Kun Li, He Mengmeng.

**Project administration:** Kun Li, Junjun Huo.

**Software:** He Mengmeng.

**Validation:** Junjun Huo.

**Visualization:** Kun Li.

**Writing – original draft:** Kun Li, He Mengmeng.

**Writing – review & editing:** Junjun Huo.

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
