## [Decision Letter · Decision Letter 0]

25 Jan 2022

PONE-D-21-33034Digital Inclusive Finance and Asset Allocation of Chinese Residents: Evidence from the China Household Finance SurveyPLOS ONE

Dear Dr. Mengmeng,

Thank you for submitting your manuscript to PLOS ONE. After careful consideration, we feel that it has merit but does not fully meet PLOS ONE’s publication criteria as it currently stands. Therefore, we invite you to submit a revised version of the manuscript that addresses the points raised during the review process.

We look forward to receiving your revised manuscript.

Kind regards,

Elisa Ughetto

Academic Editor

PLOS ONE

Journal Requirements:

Reviewers' comments:

Reviewer's Responses to Questions

**Comments to the Author**

1. Is the manuscript technically sound, and do the data support the conclusions?

Reviewer #1: Yes

2. Has the statistical analysis been performed appropriately and rigorously? 

Reviewer #1: Yes

3. Have the authors made all data underlying the findings in their manuscript fully available?

Reviewer #1: Yes

4. Is the manuscript presented in an intelligible fashion and written in standard English?

Reviewer #1: Yes

5. Review Comments to the Author

Reviewer #1: Title：Digital Inclusive Finance and Asset Allocation of Chinese Residents: Evidence from the China Household Finance Survey

This paper investigates the impact of digital inclusive finance development on the effectiveness of financial asset allocation of Chinese households using micro data from the China Household Finance Survey. The authors first propose a theoretical analysis framework and mechanism of action hypothesis, and then conduct a detailed empirical test with accurate methodology, reliable data, and convincing conclusions. The conclusions show that digital financial inclusion can significantly increase the proportion of households' risky asset allocation and promote households' reasonable participation in risky financial markets to improve the allocation efficiency of household resources. This finding has some value in developing countries such as China, and may promote the attention and application of digital inclusive finance as an emerging financial instrument in emerging economies.

Here are some comments I made for this article for the author's reference. Overall, I recommend this article to be published after minor revision.

Abstract: The abstract should be as short and concise as possible, so that the main research content and conclusions of the article can be introduced clearly, and the author is recommended to compress the number of words to within 120 words.

Introduction: The author of this section lacks the contribution of the article and its role in the real world, so I suggest the author to revise and add.

Literature review: The literature review should have a review of the literature and the marginal contribution of this article to the existing literature at the end, which is missing in this article.

Methods: The authors should have presented all the methods used in the article, including the robustness testing section.

The authors should present the Chinese Household Finance Survey, as this is the data you used.

The authors use the important variable "financial literacy", what is the appropriate way to measure this variable?

Conclusion: The conclusions and recommendations are too general, not specific and not relevant.

In addition, the authors are advised to revise the literature and format according to the journal's requirements.

6. PLOS authors have the option to publish the peer review history of their article (what does this mean?). If published, this will include your full peer review and any attached files.

Reviewer #1: No

---

## [Author Response · Author response to Decision Letter 0]

15 Mar 2022

Re: Manuscript reference No. PONE-D-21-33034

Dear editor and reviewer,

Please find attached a revised version of our manuscript “Digital Inclusive Finance and Asset Allocation of Chinese Residents: Evidence from the China Household Finance Survey?”, which we would like to resubmit for publication in Plos One.

Your comments and those of the reviewers were highly insightful and enabled us to greatly improve the quality of our manuscript. In the following pages are our point-by-point responses to each of the comments of the reviewers as well as your own comments.

Revisions in the text are shown using yellow highlight. We hope that the revisions in the manuscript and our accompanying responses will be sufficient to make our manuscript suitable for publication.

We shall look forward to hearing from you at your earliest convenience.

Responses to the comments of Reviewer #1

1.Abstract: The abstract should be as short and concise as possible, so that the main research content and conclusions of the article can be introduced clearly, and the author is recommended to compress the number of words to within 120 words.

Response: We appreciate the reviewers' comments and we have made the following changes in response to the reviewers' comments.

Abstract: Combined with the expected utility theory, this paper constructs a theoretical analysis framework including the development level, financial literacy, and intelligence level of Inclusive Finance, puts forward the hypothesis of the development of digital Inclusive Finance on household asset allocation, and uses the data of China's household finance survey to verify the theory proposed in this paper. The empirical results show that: (1) digital inclusive Finance can significantly improve the allocation proportion of household risk assets, promote the rational participation of households in the risk financial market, and improve the allocation efficiency of household resources. (2) Digital inclusive finance can significantly improve the income level of family financial investment and optimize family investment decision-making.

2.Introduction: The author of this section lacks the contribution of the article and its role in the real world, so I suggest the author to revise and add.

Response: We appreciate the reviewers' comments and we have made the following changes in response to the reviewers' comments. 

The contribution of this paper and its significance to the real world are shown in the following aspects. In terms of academic importance, the research results of this paper can supplement the existing micro-level impact research and mechanism research of digital Inclusive Finance. Based on the previous research results and chefs' data, this paper studies the impact and mechanism of the development of digital Inclusive Finance on the effectiveness of household portfolios. Compared with existing studies, this paper considers the result of the development of digital Inclusive Finance on investment returns so that this study has both investment breadth and investment depth. This paper analyzes the mechanism of the above impact and fully considers the heterogeneity of people. This paper uses municipal data, which improves the accuracy compared with previous studies using local data, making the conclusion more robust. In addition, the development of China's digital Inclusive Finance has solid national characteristics. In terms of financial services, there is a large gap between China's financial market and that of developed countries, such as insufficient coverage of traditional financial services, a large regional gap between the rich and the poor, and uneven financial literacy of residents. At the level of technological development, the progress of digital technology has made Inclusive Finance go deep into all aspects of people's daily life and closely integrate with the needs of real life. In terms of policy, China has adopted a policy of structural relaxation while strict supervision, which has brought digital inclusive Finance into the fast lane of development. The above phenomenon determines that the reference significance of foreign research results to China is limited. Therefore, it is necessary to conduct in-depth research on the impact of China's national conditions on the development of Inclusive Finance

In terms of practical significance, the research results of this paper help optimize household investment decision-making. In 2020, China's task of comprehensively eradicating poverty will be achieved, the amount of family investment will increase, and there is a massive demand for wealth management. However, the development of China's household financial market is not perfect. The "limited participation" in the traditional financial market and "excessive participation" in everyday financial need and the household portfolio lacks rationality and effectiveness. How to guide Chinese families to make rational use of the risk financial market, optimize investment decisions, increase family economic well-being, and give full play to the advantages of Inclusive Finance to improve the financial market has become a problem that can not be ignored.

In terms of policy significance, the research results of this paper will help the government formulate accurate and effective inclusive financial development policies and make practical contributions to China to seize the opportunity of targeted poverty alleviation and solve the problem of farmers' poverty. Inclusive Finance helps expand financial coverage, break through the limitations of traditional economic space, and has an essential impact on alleviating Financial Exclusion, reducing the poverty rate, and narrowing the income gap between urban and rural areas. Furthermore, through heterogeneity analysis, this paper explores the differences of different groups affected by the development of Inclusive Finance so that the government can more effectively identify financial assistance groups and improve the accuracy of China's inclusive financial services and the effectiveness of Inclusive Finance-related policies.

3.Literature review: The literature review should have a review of the literature and the marginal contribution of this article to the existing literature at the end, which is missing in this article.

Response: We appreciate the reviewers' comments and we have made the following changes in response to the reviewers' comments. 

4. Review of the literature

Foreign financial markets developed earlier, and the system is relatively perfect. Foreign scholars have more comprehensive and rich research on family financial asset allocation. There are still some problems in China's financial market compared with foreign countries, such as unreasonable investment structure and unreasonable financial product structure. In addition, there are still some problems in China, such as farmers' poverty complex and expensive financing for small enterprises. These difficulties faced by traditional Finance provide space for developing Inclusive Finance in China. Compared with family finance, inclusive Finance has been created for a short time,

At present, the empirical research on Inclusive Finance in China mainly focuses on the macro impact of Inclusive Finance, such as promoting economic growth and narrowing the income gap between urban and rural areas. The main research body is mainly traditional financial institutions (banks, insurance companies, etc.). On the other hand, foreign research covers a wide range, primarily focusing on the fact that inclusive Finance can reduce financial exclusion and eliminate economic discrimination, which impacts economic development and social stability.

Although scholars have conducted various studies on the impact of Inclusive Finance, there are still the following deficiencies:

(1) Existing studies generally pay attention to the macro impact of the development of Inclusive Finance, less research on the micro implications of Inclusive Finance, and less empirical literature on the relationship between digital finance development and residents' financial management and investment. According to the statistics of Bain consulting in 2018, China has more than 190 trillion yuan of personal investable assets and has a massive demand for wealth management. However, China's household asset allocation is mainly bank deposits and real estate, which lacks rationality and effectiveness. Therefore, this paper believes that the research on the impact of the development of digital Inclusive Finance on household financial asset allocation is fundamental.

(2) Most of the existing empirical studies are from measurement but fail to clarify the transmission mechanism of inclusive financial development to family economic behavior. This paper believes that defining the impact mechanism of Inclusive Finance and its impact on different groups will help to formulate targeted monetary policies, help China improve the financial service system, and enhance the degree of financial inclusion in China.

(3) Foreign research covers a wide range. However, due to the rapid development of the Internet in recent years, there is a large gap between China's financial market and developed countries. Furthermore, due to the insufficient coverage of traditional financial services, the large gap between the rich and the poor in China, the uneven financial literacy of residents, and other factors, the reference significance of foreign research results to China is limited; it is necessary to conduct in-depth research on the impact of China's national conditions on the development of Inclusive Finance.

4.Methods: The authors should have presented all the methods used in the article, including the robustness testing section.

The authors should present the Chinese Household Finance Survey, as this is the data you used.

The authors use the important variable "financial literacy", what is the appropriate way to measure this variable?

Response: We appreciate the reviewers' comments and we have made the following changes in response to the reviewers' comments. 

1. Data source

China Household Finance Survey (CHFS) is a nationwide sampling project on micro information of household finance. The database adopts a stratified, three-stage, and PPS sampling method, with a low rejection rate and data close to the census results, which is representative and can facilitate research on household financial behavior and macro policy formulation in China. The CHFS data covers information on household housing assets, financial finances, liabilities and credit constraints, income consumption, social security, payment habits, and demographic characteristics, providing a comprehensive and detailed picture of household economic behavior.

 The CHFS database is open for applications for 2011, 2013, 2015, 2017, and 2019 survey results data. This paper uses data from five rounds of survey results from 2011-2019, whose sample covers 29 provinces (including autonomous regions and municipalities directly under the central government), 355 counties (including districts and county-level cities), and 1428 village (residence) committees, with a total sample of 140011 households. Since the original CHFS data contained a large number of missing values and outliers (e.g., household income, household assets, personal risk attitudes, etc.), to ensure the reliability of the results, this paper eliminated samples containing missing values and outliers for critical variables (e.g., household income, household assets, etc.), and obtained a total of 118,784 valid samples.

 Financial literacy is a combination of financial knowledge, behaviours and skills possessed by an individual, and measures the financial knowledge reserve possessed by an individual and the ability to apply financial knowledge to solve practical financial problems. This paper uses respondents' responses to the financial literacy questions in the questionnaire as the basis for determining the level of financial literacy.

3. Model construction

In studying the impact of the development of inclusive digital Finance on the proportion of household financial risk holdings, this paper selects the proportion of household risky financial assets holdings finance to total assets as the explanatory variable. Since the ratio of risky investments is mainly a 0-1 variable, this paper uses the Tobit model for estimation, and the model is set as follows.

(4)

where represents the local financial inclusion index and defines the control variables, including individual, household, and city-level characteristics.

In studying the impact of digital financial inclusion development on household investment returns, the Sharpe ratio is chosen as the explanatory variable. However, for households that do not participate in investment in risky financial assets, their corresponding Sharpe ratios are 0. Therefore, Sharpe ratios are mostly 0-1 variables, which are also estimated using Tobit models, and the related regression models are as follows.

(6)

This paper argues that digital financial inclusion can affect household economic behavior by influencing the level of financial literacy and the level of Internet usage. The analysis for Mechanism 1 can be divided into two steps: the first step tests the impact of financial inclusion development on financial literacy, and the corresponding regression model follows.

(7)

To fully consider the population heterogeneity, this paper adds the asset interaction term and the age group interaction term to the above model to explore the differences in the impact of digital financial inclusion development on the financial literacy of different population groups. The corresponding regression models are as follows.

(8)

(9)

The analysis idea for mechanism two also adopts a similar concept to the analysis of mechanism one. This paper chooses two variables, whether the household uses Internet financial products (using Internet financial products takes the value of 1. Otherwise, it takes the value of 0) and the household's use of Internet (used takes the value of 1. Otherwise, it takes the value of 0), as the willingness to use Internet variables for regression analysis. The corresponding regression models are as follows.

(10)

(11)

(12)

5.Conclusion: The conclusions and recommendations are too general, not specific and not relevant.

Response: We appreciate the reviewers' comments and we have made the following changes in response to the reviewers' comments. 

VII. Conclusions and Recommendations

1. Conclusion

This paper studies the impact of the development of digital Inclusive Finance on the effectiveness of household financial asset allocation. Firstly, this paper combs the measurement standards of relevant variables, the relevant research results of Inclusive Finance and household asset allocation, and finds that the current research on the household micro impact of digital Inclusive Finance mostly stays at the level of household asset types and the proportion of household risk assets, without considering the return on household investment; Most of the existing empirical studies fail to clarify the transmission mechanism of inclusive financial development to household financial investment behavior from the perspective of measurement. This study aims to improve the above deficiencies, supplement and improve the existing research results.

This paper combines utility theory to construct a theoretical analysis framework that includes the level of financial inclusion development, financial literacy, and intelligence, and then derives and analyzes a hypothesis about the impact and mechanism of action between the level of digital financial inclusion development and household asset allocation. This paper argues that the development of inclusive digital Finance can enhance the effectiveness of household asset allocation, and this effect is carried out by improving financial literacy and residents' willingness to use the Internet. The mechanism of action differs among different income and age groups, and the effect is more pronounced in older age groups with lower asset levels.

 In terms of empirical evidence, this paper defines the effectiveness of household asset allocation in terms of two dimensions: household risk market participation and investment returns, and studies investment breadth and depth. This paper selects CHFS data and BYU Digital Inclusive Finance Index to test the hypotheses proposed in this paper. The empirical results show that (1) the development of inclusive digital Finance can significantly increase the proportion of household risky asset allocation and promote households' reasonable participation in the complex financial market to improve the allocation efficiency of household resources. (2) The development of inclusive digital Finance can significantly improve the return level of household financial investment and optimize household investment decisions. Finally, this paper analyzes the mechanism of the above findings. It concludes that: (1) the development of inclusive digital Finance can improve residents' financial literacy, enhance households' knowledge about financial product selection and financial risk management, improve the efficiency of financial decision-making, and thus influence households' financial behavior. And the mechanism has a more substantial impact on people with lower income and older age. (2) The development of digital inclusive financial development can fully play the advantages of Internet technology, increase residents' willingness to use the Internet, and enable them to enjoy the dividends of technological development, thus optimizing household financial market participation.

2. Recommendation

Based on the above research findings, this paper proposes the following policy recommendations.

 (1) Play a leading role of the government and build a suitable environment for developing inclusive digital Finance. Digital inclusive Finance can break the current financial dilemma, help implement the rural revitalization strategy, and realize the leap from precise poverty alleviation and poverty eradication to precise policy-making. However, the development of inclusive digital Finance needs to rely on a good development environment. Therefore, the government should focus on improving the infrastructure construction and credit system to develop digital inclusive Finance to promote the complementary advantages of digital inclusive finance-related parties and form an excellent financial, ecological pattern of win-win cooperation.

 (2) Focusing on financial information regulation and regulating online financial information delivery. This study considers improving financial literacy as one of the mechanisms of digital inclusive Finance's influence on household economic behavior. However, there are still problems in China, such as spreading false financial information, promoting financial products and services banned by relevant departments, and distorting the interpretation of national fiscal and monetary policies and financial management policies, which hinder residents' accurate access to practical information. Therefore, the government should focus on improving the quality of financial information services, punishing illegal and unlawful dissemination of false financial information and wrong financial information, and promoting the orderly development of financial information services.

 (3) Strengthen financial education and enhance household financial literacy. This paper finds that improving the level of household financial literacy can significantly enhance the effectiveness of household asset allocation. At present, the overall financial literacy of Chinese residents is at a low level. As a result, financial frauds occur occasionally, and most residents are unable to correctly understand financial products and correctly view financial risks. Therefore, the government should focus on increasing financial education, popularizing basic financial knowledge through various channels such as the Internet and television, helping residents establish risk awareness, guiding investors to invest rationally, and promoting the healthy and sustainable development of China's financial market.

(4) Give full play to the advantages of advanced technology and standardize and promote financial service innovation under the condition of controllable risk. Financial service innovation can enrich product supply, optimize service quality and improve financial service efficiency. The government should implement the new development concept, comply with the development requirements of the times, and strive to promote the deep integration of Finance and science and technology under the condition of controllable risk, so as to make the achievements of scientific and technological development better benefit the people's livelihood.

---

## [Decision Letter · Decision Letter 1]

1 Apr 2022

Digital Inclusive Finance and Asset Allocation of Chinese Residents: Evidence from the China Household Finance Survey

PONE-D-21-33034R1

Dear Dr. Li,

We’re pleased to inform you that your manuscript has been judged scientifically suitable for publication and will be formally accepted for publication once it meets all outstanding technical requirements.

Kind regards,

Elisa Ughetto

Academic Editor

PLOS ONE

Additional Editor Comments (optional):

Reviewers' comments:

Reviewer's Responses to Questions

**Comments to the Author**

1. If the authors have adequately addressed your comments raised in a previous round of review and you feel that this manuscript is now acceptable for publication, you may indicate that here to bypass the “Comments to the Author” section, enter your conflict of interest statement in the “Confidential to Editor” section, and submit your "Accept" recommendation.

Reviewer #1: All comments have been addressed

2. Is the manuscript technically sound, and do the data support the conclusions?

Reviewer #1: Yes

3. Has the statistical analysis been performed appropriately and rigorously? 

Reviewer #1: Yes

4. Have the authors made all data underlying the findings in their manuscript fully available?

Reviewer #1: Yes

5. Is the manuscript presented in an intelligible fashion and written in standard English?

Reviewer #1: Yes

6. Review Comments to the Author

Reviewer #1: The author has fully considered my comments. All comments have been addressed and are recommended for publication.

7. PLOS authors have the option to publish the peer review history of their article (what does this mean?). If published, this will include your full peer review and any attached files.

Reviewer #1: No

---

## [Editor Report · Acceptance letter]

14 Apr 2022

PONE-D-21-33034R1 

Digital Inclusive Finance and Asset Allocation of Chinese Residents: Evidence from the China Household Finance Survey 

Dear Dr. Li:

I'm pleased to inform you that your manuscript has been deemed suitable for publication in PLOS ONE. Congratulations! Your manuscript is now with our production department. 

Kind regards, 

on behalf of

Prof. Elisa Ughetto 

Academic Editor

PLOS ONE